computational biology/health and disease and epidemiology

SARS-CoV-2, social-ecological system, compartmental model, economic pressure

**Author for correspondence:**
I. Santamaría-Holek
e-mail: isholek.fc@gmail.com

# Possible fates of the spread of SARS-CoV-2 in the Mexican context

I. Santamaría-Holek[1] and V. Castaño[2]

[1]UMDI-J, Facultad de Ciencias, and [2]Centro de Física Aplicada y Tecnología Avanzada, Universidad Nacional Autónoma de México, Juriquilla, Querétaro CP 76230, México

  IS-H, 0000-0002-5306-197X; VC, 0000-0002-2983-5293

The determination of the adequate time for house confinement and when social distancing restrictions should end are now two of the main challenges that any country has to face in an ongoing battle against SARS-CoV-2. The possibility of a new outbreak of the pandemic and how to avoid it is, nowadays, one of the primary objectives of epidemiological research. In this work, we present an innovative compartmental model that explicitly introduces the number of active cases, and employ it as a conceptual tool to explore the possible fates of the spread of SARS-CoV-2 in the Mexican context. We incorporated the impact of starting, inattention and end of restrictive social policies on the pandemic's time evolution via time-dependent corrections to the infection rates. The magnitude and impact on the epidemic due to post-social restrictive policies are also studied. The scenarios generated by the model could help authorities determine an adequate time and population load that may be allowed to reassume normal activities.

## 1. Introduction

Humankind and environment are bidirectionally linked through a dynamical hierarchical feedback [1–6]. Society impacts the natural systems by exploiting, extracting, and (sometimes) restoring resources, as well as polluting. The environment affects social networks by providing or restricting the resources required for their survival [7]. If the relation between both systems implies that their impacts do not threaten their existence, the relationship is sustainable [6,8].

Overpopulation, overconsumption and economic inequality have decreased the stability of social-ecological systems. This fact leads to possible unsustainable future scenarios and risks the viability of the roles and functions of both human and natural systems [3–5].

The actual SARS-CoV-2 pandemic is an excellent example of how a natural event can break this coupling among societies and the environment. Historically, many civilizations have

collapsed due to their inadequate resource-management policies and the lack of an ability to modify their cultural practices, like the consumption of non-traditional foods [9,10], thereby creating unsustainable trajectories [5]. The beginnings of pandemics often result from the transgression of animal habitats and leads to severe disruption and damage to our lives in both the public health and economic domains [6,11,12].

Given the global character of SARS-CoV-2 spread, innovative epidemiological analyses have been recently proposed, including complex network analysis and conceptual maps, as well as compartmental models specifically created for the COVID-19 evolution, e.g. [13–21], among many others. Compartmental models are valuable since they can address important aspects of the spread of the disease by analysing the effects that social distancing and domestic confinement or pharmaceutical interventions may have on the evolution of the pandemics [14,17,19–21]. These models are of a fundamental importance since they attempt to correlate the clinical aspects of the COVID-19 disease [15–21] with the spreading rates of the SARS-CoV-2 and the contact probabilities that characterize the interaction among different population sectors, as well as other sources of infection, such as direct contact with contaminated surfaces or air transmission [19]. They are also very important in virology and in epidemiology to control outbreaks, invasions and transmissions [10,22–26], and also for informing and analysing health policies [13,24,26–28].

Several compartmental models about the spread of the SARS-CoV-2 pandemic have been proposed in the past few months [15–21]. Their contribution to the establishment of the correlations between clinical observations and the transmission rates is valuable and represents a solid basis for further development. For example, the study reported in [16,17] simulated the Wuhan outbreak by using a deterministic SEIR model over the course of a year. This study assumed the city of Wuhan as a closed system with a constant population of 11 million. The population, divided according to usual SEIR compartments, included asymptomatic and subclinical cases, with the addition of evaluating the evolution from the age 5 to age of 75 in 5-year bins. The study analysed the adequate time for domestic confinement ending in terms of the median number of infections for the different sectors and age distributions. These interesting projections pointed out the danger of a premature lifting of interventions, which would eventually lead to a second peak of infection.

However, the dispersion of the virus over countries having very different health, social-economical and cultural conditions, makes necessary the formulation of specific models for accounting the many local dynamics of the pandemic. In this work, we study the case of Mexico due to the peculiarities shown in the behaviour of the pandemic and the lack of extensive testing for early detection of cases. The challenge has already been faced by different local groups that produced different models accounting for different aspects of the initial stages of the pandemic, e.g. [15,19]. These models considered different possible routes of infection and evaluated the effect of social policies on social distancing and home confinement, allowing to test the universality and representativity of the models.

Accordingly, based on the latest specialized reports on mathematical modelling, our present work uses explicit time dependence of the transmission rates in the force-of-infection terms, as well as the introduction of other communities of susceptible individuals, to fit the most basic trends of the pandemics that are daily reported by the health authorities. These data correspond to the cumulative number of confirmed cases, the number of deaths and recovered individuals. From these data, two crucial information sub-products are deduced: the daily cases and the number of individuals that are not considered infectious and have not yet been reported as recovered or dead. In a similar form to that described in the previously cited models, our approach presents a perspective on the effects of the health policies implemented on the different sectors and the effect of the time dependence of the social distancing and house confinement measures. Our modelling strategy has some similitudes with SEIR (susceptible, exposed, infectious, recovered) and SIRD (susceptible, infectious, recovered, deceased) models. However, it shares some aspects with more recent strategies that incorporate the significant fact that the transmission coefficients are time-dependent [15–17,19,20]. Innovations in the present work come from the definition of three compartments that do not coincide with classical compartmental model definitions and that account for the lack of information due to the non-application of extensive tests for determining the SARS-CoV-2 spread in Mexico. Far beyond characterizing the effect of social distancing and house confinement measures, we also look for the effect of the inattention of the health measures due to economic pressure, which recently increased the number of contagions. Educational campaigns are helpful but find resistances related to the very non-homogeneous cultural, social and epistemological characteristics of the Mexican population. The technical details of the differences and similarities of our approach with previous reports will be discussed in what follows, emphasizing the need for the modification.

## 2. The modelling mechanism and the evolution equations

Our model aims to fit and extrapolate to later times the evolution of the essential data reported by the health authorities on cumulative confirmed cases, deaths and recovered individuals. From these three sets of data, we derive the information for the daily cases, and a class of individuals not yet recovered or dead and classified as active cases in [29]. The model also aims to reproduce quantitatively these data sub-products, which are essential for elaborating health policies. To achieve our objectives, we propose a variation of the classical SEIR and SIRD models that shares some similitudes with the models on SARS-CoV-2 spread we have discussed in the introduction, e.g. [16,17,19,20]. The variations introduced were designed to improve our understanding and projections of the possible future scenarios of the pandemic depending on governmental decisions as to the duration of the social distancing and domestic confinement, as well as the importance of restrictions (or their lack) imposed on society after ending lockdown.

In the formulation of the model, we consider some specific characteristics of the spread of the SARS-CoV-2 virus among humans. The parameters of the model were initially estimated in agreement with previous models and clinical observations reported in the literature [15–21]. However, their values were improved by performing fits of the data reported by the health authorities. In the first approach, we have used the information available for the Mexican scenario to the date of 19 May. Later, we updated the data and actualized the parameters of the model to the date of 7 August.

We consider that there are three forms by which a susceptible individual ($S$) becomes infected. The first one that dominates at the early stages of the outbreak is closely related with the direct physical contact with the virus ($V$) through, for example, contaminated surfaces in public markets or public transport. The result of these early-stage exposures to the virus is the appearance of individuals without knowledge of their own condition. It is well established that an essential aspect of the spread of SARS-CoV-2 in the population is the high proportion of pre-symptomatic and asymptomatic [16,19,20] individuals that may infect vulnerable sectors without being aware of the situation. Most of these individuals are unreported cases that play a determinant role in the spread of the virus and, together with the unreported symptomatic cases, should be considered in the dynamics of a separated compartment $U$. Therefore, the early stages of the pandemic can be described by the reaction

$$S + V \xrightarrow{k_v} U. \tag{2.1}$$

In a first approximation, the amount of virus $V$ can be considered constant, and $k_v$, the free-virus transmission rate, refers to the number of contacts per number of susceptible individuals with predominant non-human agents at the early stages of the pandemic. This assumption on the early stages of evolution of the pandemic is in agreement with previous reports where an initial exponential growth of the confirmed cases was assumed; see [20]. The $k_v$ parameter is important because the number of individuals that may have direct contact with non-human virus sources is much less than the total population of a country and grows with the total number of susceptible individuals $S_0$: $k_v = \tilde{\beta} S_0$, where $\tilde{\beta}$ is a contact probability per unit time. The implementation of a similar unreported compartment was considered in [20].

Previous approaches have postulated an evolution equation for the free-virus $V$ that depends on other compartments associated with infected and exposed individuals, e.g. [19]. Although this procedure can provide a more detailed vision for the contagions with the free-virus, it has the disadvantage that it requires new parameters for its characterization. In our approximation, the exposure to the free-virus is established via a single relation between the virus and the susceptible population. Therefore, on average, it can be measured in terms of the number of direct contacts with non-human sources.

Consistent with the previous discussion, we will assume that the pandemic's initial delay is not determined by the existence of the classical exposed compartment but by reaction (2.1). Hence, in the second stage of evolution, the rise of unreported infected individuals enhances the infections by introducing nonlinearities in the evolution of the pandemic. This process is represented by the reaction

$$S + U \xrightarrow{k_e} 2U, \tag{2.2}$$

where the transmission rate $k_e$ measures the number of contacts per day between susceptible and unreported infected individuals. Consistent with compartmental model theories, we assume that $k_e$ is implicitly proportional to a contact probability per unit time and inversely proportional to a population number. In our approach, we will consider that this population is an estimate of total susceptible individuals $S_0$: $k_e \simeq \beta / S_0$. More fundamental approaches would aim to determine the value of these

parameters using several techniques [15–21]. Here, we used those values as starting ones but improved them through the fits of the data. We note that those parameters may be influenced by pharmacological treatments from country to country and on different conditions of the health-care services.

The third infection possibility relates to the fact that a proportion of the pre-symptomatic and asymptomatic individuals develop symptoms and go to health services, thus becoming reported infected individuals ($I$). Hence, the infection of susceptible individuals by this component is characterized by

$$S + I \xrightarrow{k_i} U + I, \tag{2.3}$$

where the transmission rate $k_i$ measures the number of contacts per day between susceptible and reported infected individuals. In similar form to $k_e$, we have $k_i \simeq \hat{\beta}/S_0$ with the corresponding contact probability per unit time $\hat{\beta}$. It is expected that $k_e > k_i$. These three contagion reactions suppose that the triggering mechanism yields predominantly pre-symptomatic and asymptomatic non-reported individuals until symptoms appear. This assumption may be validated in countries where massive SARS-CoV-2 tests are applied to the population independently of the manifestation, or not, of symptoms.

The successive evolution of the dynamics is the transition of individuals from $U$ and $I$ components towards the recovered ($R$) and death ($D$) compartments.

The first transition we consider in this third stage is related to the compartment of unreported individuals. We have already explained that this compartment does not correspond to the usual definition of exposed individuals that may, or may not, develop symptoms, but with the number of individuals (exposed, pre-symptomatic, asymptomatic and even symptomatic) not officially registered. The consequence of under-registration is that individuals of the compartment $U$ that become recovered or die are neither reported in the statistics. Given this fact, we are compelled to assume that the unreported individuals only leave their group when they pass to the (reported) infected compartment and that the recovered and the dead individuals (that are reported) originate only from the compartment $I$. Therefore, the evolution of the unreported individuals occurs according to

$$U \xrightarrow{\alpha} I, \tag{2.4}$$

where the inverse of $\alpha$ measures the time of an unreported individual to start presenting symptoms, that is, between 2 and 14 days with median $\approx 11$ days [30].

In the following, we will consider that the compartment $I$ evolves towards three subpopulations corresponding to recovered ($R$), death ($D$) and pre-recovered active ($A$) individuals. As far as we know, no pre-recovered sector is introduced in compartmental models of other diseases or previous models for the SARS-CoV-2 spread. However, we consider it necessary to introduce it since it reflects that, on 27 May 2020, the WHO actualized the criteria for discharging patients from isolation, because many patients whose symptoms have resolved may still test positive for the SARS-CoV-2 virus by reverse transcription–polymerase chain reaction (RT-PCR) for many weeks. These pre-recovered active patients are not likely to be infectious but are still infected, and hence, they are not part of the infection force. Hence, the reactions representing the leave of the compartment $I$ comprise

$$r I \xrightarrow{\gamma} R, \tag{2.5}$$

$$d I \xrightarrow{\delta} D, \tag{2.6}$$

$$a I \xrightarrow{\epsilon} A. \tag{2.7}$$

Here, the parameters $r$, $d$, $a$ measure the proportions of individuals that recover, die, or are pre-recovered from the total number of known infected individuals, and satisfy the condition $r + d + a = 1$. Mathematically, it is equivalent to avoid these parameters and only change the rate constants appropriately during the fit of the data. However, their introduction helps maintain the stoichiometry of the compartment $I$. Additionally, they keep unaltered the dependence of the rates on the intrinsic interaction of single individuals with the virus [15–21].

The parameter $\gamma^{-1}$ measures the characteristic time after the which the symptoms from a seriously ill individual disappear (between three to six weeks), and $\delta^{-1}$ is the characteristic time endured from the beginning of symptoms until death (which ranges between 17 and 21 days) [16,17,20,22,28,31–33]. The parameter $\epsilon^{-1}$ is the period during which an infected individual maintains manifests active symptoms, and thus it will be considered similar to $\alpha^{-1}$. This assumption is validated, a posteriori, by the fact that five fits are performed simultaneously. These are the total number of individuals of infected, recovered, dead sectors together with the number of pre-recovered and daily cases. Note that the parameters $\alpha$, $\beta$, $\gamma$ and $\epsilon$ depend mostly on features of the virus in humans and not on social contact or hygiene.

Equation (2.7) introduces an additional delay in the evolution of the epidemic that leads to the following two final steps of the dynamics

$$p\,A \xrightarrow{\delta} D \tag{2.8}$$

and

$$(1-p)\,A \xrightarrow{\gamma} R, \tag{2.9}$$

since a pre-recovered individual has only two options; it can recover or die. In these last reactions, we have assumed that the recovering and death rate constants are the same as in the case of the infected sector. This assumption can be relaxed accordingly with the characteristics of the data compiled for the disease. We have also introduced the statistical parameter $p$ which, as mentioned before, could depend on the quality of the health care. In Mexico, according to the official reports, this proportion ranges from 9% to 12% of the known infected population [34]. Therefore, the parameter will take values in the range $p \approx 0.09$–$0.12$.

Reactions (2.1) to (2.9) yield the following set of nonlinear differential equations

$$\frac{dS}{dt} = \mu S - (k_v + k_e U + k_i I)S, \tag{2.10}$$

$$\frac{dU}{dt} = (k_v + k_e U + k_i I)S - \alpha U, \tag{2.11}$$

$$\frac{dI}{dt} = \alpha U - (\gamma r + \delta d + \epsilon a)I, \tag{2.12}$$

$$\frac{dA}{dt} = \epsilon a I - [\delta p + \gamma(1-p)]A, \tag{2.13}$$

$$\frac{dR}{dt} = \gamma r I + \gamma(1-p)\,A, \tag{2.14}$$

$$\frac{dD}{dt} = \delta d I + \delta p\,A. \tag{2.15}$$

In this mechanism, the total count of all members of all groups $N = S + U + I + A + R + D$ obeys $dN/dt = \mu S$, where $\mu$ is the exponential growth constant (rate of births minus deaths) of Mexican population before the entrance of the virus. Since we are interested only in the effects of the pandemic event we will assume without loss of generality that $N(t) = N_0$ is constant ($\mu = 0$).

In a first assessment, the number of individuals susceptible to being infected can be estimated after recognizing that approximately 75–78% of the population is urban (cities having 2500 individuals or more) [35], thus leading to a total urban population about 101.4 million. In agreement with statistical data from European countries (up to 19 May), about 0.3% was susceptible to being infected from this urban population. This number of susceptible individuals is estimated at around $S_0 \approx 305\,000$ individuals and was consistent with the more generous estimates of the pandemic's impact in Mexico. Those estimates led to the idea that the pandemic could be controlled in a few urban centres. However, a more realistic assessment is obtained after the subsequent evolution of the pandemic (up to 7 August). We consider that keeping both assessments and their results to which they lead is illustrative of the weakness of using purely statistical methods for the initial estimation of parameters and populations of the sectors relevant to the model.

# 3. Results

The data reported by the authorities are (i) the (cumulative) number of confirmed cases, (ii) the number of recovered individuals, and (iii) the number of deaths. From these data, one may derive (iv) the number of daily cases, and (v) the number of pre-recovered active cases. Since no extensive testing is applied to the population, the information about pre-symptomatic and asymptomatic individuals is not available. Thus, the cumulative number of infected individuals must be fitted using the number of pre-recovered active cases, which pertain to the reported class.

## 3.1. Fit criteria

To be reliable, the fit of data for the total number of confirmed infected individuals, Confirmed($t$), has to be simultaneous with the fit of the number of recovered and dead individuals, as well as with the daily

number of infected individuals, Day($t$), and the number of pre-recovered active cases, $A(t)$. That is, for reliable predictions from the model, these five fits have to be performed simultaneously.

The total number Confirmed($t$) of confirmed infected individuals as a function of time is given by the sum of the three sectors entering into relations (2.7)–(2.9), that is

$$\text{Confirmed}(t) = R(t) + D(t) + A(t), \tag{3.1}$$

which is obtained by a simple conservation of individuals. Note that this definition is equivalent to that used in [20] since it coincides with the time integral of the loss term in equation (2.12).

The number of daily infected individuals is proportional to the cumulative number of infected individuals multiplied by the inverse characteristic times of those becoming recovered, death and pre-recovered, that is

$$\text{Day}(t) = (\gamma r + \delta g + \epsilon a) \int_{1}^{t} [I(t') - I(t'-1)]\mathrm{d}t', \tag{3.2}$$

which is the loss term of equation (2.12).

## 3.2. Results

The set of equations (2.12)–(2.15), together with the definitions (3.1) and (3.2), were used to fit the data reported by the Mexican authorities on the number of confirmed cases (red circles), recovered (blue squares) and deaths (orange triangles) in figure 1$a$; see [34,36] for details.

The fits were performed assuming that the transmission rates $k_e$ and $k_i$ are time-dependent functions since they may be affected by social distancing, domestic confinement attention and inattention, and other hygienic circumstances and policies [14,16,17,19–22]. The interventions will change the value of the transmission rate in the following form:

$$k_j^0(t) = k_j - \Delta k_j^{di}\,\theta(t - t_i) + \Delta k_j^{ep}\,\theta(t - t_{ep}) + \Delta k_j^{f}\,\theta(t - t_f), \tag{3.3}$$

where $k_j$ is the infection rate and $\theta(t - t_k)$ (with $j = i, ep$) is the Heaviside step function centred at the initial time of the domestic confinement $t_i$ and social distancing measures [$\theta(t - t_i)$]. This term is negative because these measures decrease the infection rate. The third term at the right-hand side is positive since it represents the increase of the infection rate due to economic inattention, which takes place at $t_{ep}$ [$\theta(t - t_{ep})$] before ending the lockdown. The last term corresponds to the contagion rate's increase after ending the lockdown, which takes place at $t_f$. The parameters $\Delta k_j^{di}$, $\Delta k_j^{ep}$ and $\Delta k_j^{f}$ measure the fall and rise of each transmission rate due to the start of house confinement, its inattention and the end of the confinement measures.

Simultaneously, we used definitions (3.1) and (3.2) together with the reported data to fit the number of pre-recovered active cases and the daily number of infected individuals, figure 1$c$. In figure 1$a$,$b$, the lines are the fits of our model for the Confirmed (black), recovered (blue), death (orange), unreported (orange-dashed) and infected (red). Figure 1$b$ shows the long-term projection of the model after assuming that the number of susceptible individuals was about 61% of the initially estimated number ($S_0 = 187\,000$). Simultaneously, we have make fits of figure 1$c$,$d$, that report the number of pre-recovered active cases (blue dashed line) and the number of daily cases (black line). This estimation was adopted after considering the authorities' announcements about the tendencies of several factors indicating that the maximum of the infected individuals should occur about the middle days of May. It is now evident that this estimation under-represented the situation in Mexico. A model actualization is presented below, where better estimations yield more realistic projections of the pandemic's evolution. A note in favour of the values used here is that they allowed fitting to more data concerning the evolution of the pandemic. They adequately reflect the first social distancing measures and the public inattention of those measures due to economic pressure.

According to this initial prognosis, the maximum of the infection curve could takes place around the days 75–85, corresponding to days between 12 and 22 May. The projection suggests that the 97% point of the pandemic is reached (with the social distancing and house confinement infection rates estimated before day $90 \approx 1$ June) about the week of 1–7 November (days 246–252). The number of new daily cases and pre-recovered active cases have their maxima about days 80 (19 May) and 125 (3 July), respectively.

The estimation of the infection rates for unreported $k_e$ and infected $k_i$ sectors assumes that they have to be $10^2$–$10^3$ times the inverse of the Mexican population ($7.7 \times 10^{-9}$), that is, they must lie in the range

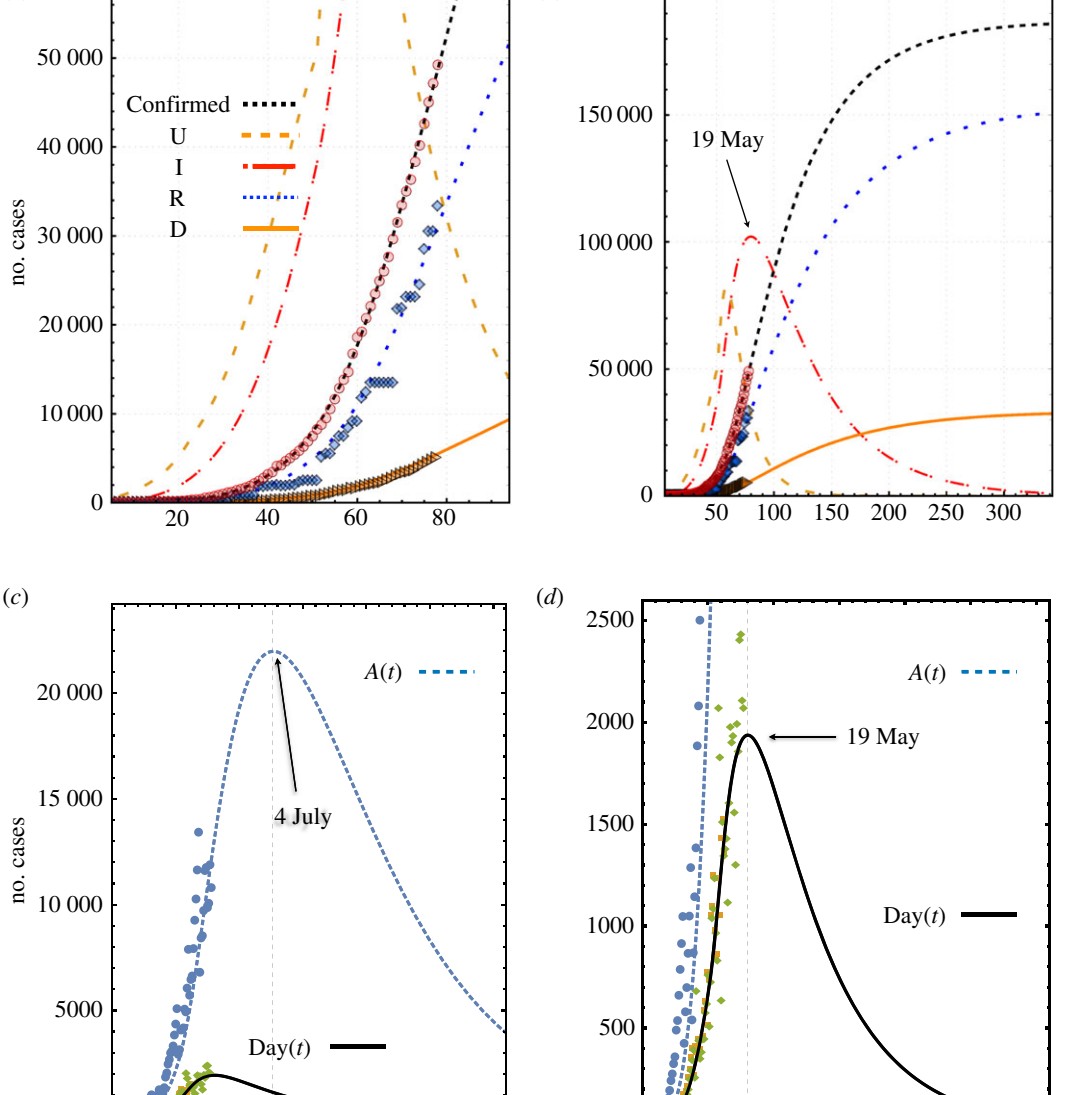

**Figure 1.** Fit of the data reported by the Mexican authorities (up to 19 May). (*a*) The symbols represent infected (red circles), recovered (blue squares) and death (orange triangles) individuals, the lines are the numerical solutions of equations (2.10)–(2.15) for $I(t)$ (red), $R(t)$ (blue), $D(t)$ (orange) and unreported $U(t)$ (orange-dashed). The black dashed line is the number of confirmed cases given by equation (3.1). (*b*) Long-term projection of the fit. (*c*) Fit of the pre-recovered active cases and (*d*) of the cumulative number of daily cases Day(*t*). The calculation of the data represented by the symbols was done using equation (3.2) (green diamonds squares) and equation (3.1) (blue circles). In the case of daily cases (orange squares), we also used data from Johns Hopkins COVID-19 global map to corroborate that our determination of the daily cases coincides with the data reported in [34].

$10^{-7}$–$10^{-6}$ days$^{-1}$ [15]. The rate at which symptoms appear is estimated to be in the interval of $10^{-2}$–$10^{-1}$ days$^{-1}$, whereas the rate at which symptoms resolve, $\gamma$, was assessed on a wide range around $10^{-2}$ days$^{-1}$, both consistent with clinical observations [31–33]. The free-virus transmission rate $k_v \sim 2.6 \times 10^{-4}$ was obtained from the fit of the early exponential growth of the cumulative number of cases until day 20. This estimation is essential to determine the initial slope of the curve of the cumulative number of cases.

Figure 2*a* shows details of the time evolution of the pre-recovered active and daily cumulative cases $A(t)$ and Day(*t*), respectively, as a function of time and compared with the data reported. It is indicated the day at which the domestic confinement measures were initiated, day 23 (23 March), and the day at which many people ended confinement to their homes due to economic pressure, day 50 (19 April),

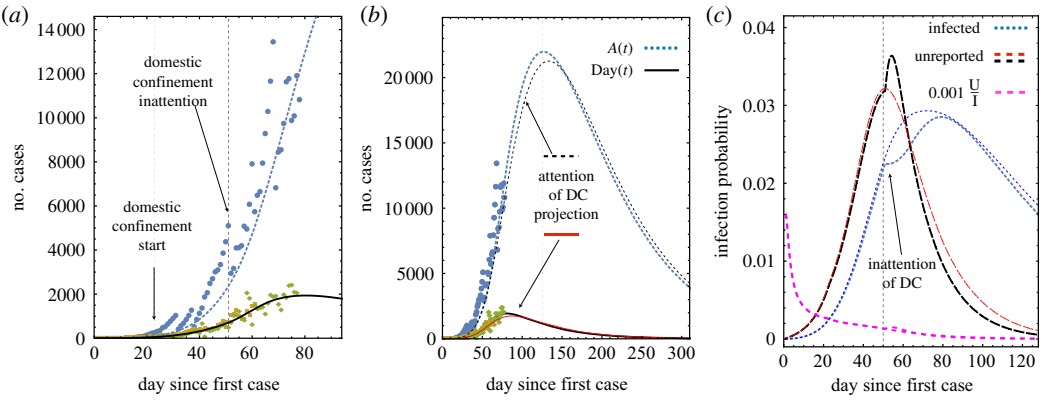

**Figure 2.** Fit of the data for the pre-recovered and daily number of cases reported by the Mexican authorities (up to 19 May) [34,36]. (a) The symbols represent pre-recovered active (blue circles) and daily cases (green and orange diamonds). The lines are the results for these two populations as obtained from numerical solutions of equations (2.10)–(2.15) for $A(t)$ (blue) and $Day(t)$ (black). (b) Comparison of the evolution of $A(t)$ and $Day(t)$ for the attention of domestic confinement (DC) (black lines) and inattention of DC (blue and red lines, respectively). (c) Infection probability rate (equations (3.8) and (3.9)) as a function of time for the unreported and infected individuals, and scaled proportion of the number of unreported individuals concerning the number of infected ones.

leading to a weakening of those measures. These measures were accounted for by introducing the corrections explained in equation (3.3) into equations (2.10) and (2.11)

$$\frac{dS}{dt} = \mu S - [k_v V + \tilde{k}_e(t)U + \tilde{k}_i(t)I]S, \tag{3.4}$$

$$\frac{dU}{dt} = [k_v V + \tilde{k}_e(t)U + \tilde{k}_i(t)I]S - \alpha U, \tag{3.5}$$

where

$$\tilde{k}_e(t) = k_e - \Delta k_e^{di}\theta(t - t_i) + \Delta k_e^{ep}\theta(t - t_{ep}) \tag{3.6}$$

and

$$\tilde{k}_i(t) = k_e - \Delta k_i^{di}\theta(t - t_i) + \Delta k_i^{ep}\theta(t - t_{ep}). \tag{3.7}$$

The definitions of $\Delta k_i^{di}$ and $\Delta k_i^{ep}$ are given after equation (3.3).

The implementation of social distancing and domestic confinement led, in the first stage, to a reduction of 11% of the infection rate of both infected and unreported individuals ($\Delta k_i^{di} \simeq 7.15 \times 10^{-8}\,\text{s}^{-1}$). However, by 19 April, circumstances associated with economic pressure (survival) produced a significant increase in the number of daily cases due to the rise in the infection rate of about 70%. This increase is evident when comparing with the case of the prediction with social distancing methods intact, see figure 2b, black dashed and solid black lines, from where it is clear that an increase of about 5% of the active and daily cases took place. The dashed blue and red lines are the predictions under the assumption that social distancing and domestic confinement were respected. The blue line corresponds to an increase of the infection rate of $\Delta k_i^{ep} \simeq 4.6 \times 10^{-7}\,\text{s}^{-1}$. It entered into play in the time evolution of the model on 19 April (day 50). This small increase considerably improved the fit of the infection data in figure 1 and is responsible for the increase of the pre-recovered active and daily cases. The accuracy of the prediction in the last stage is because the reporting of the number of individuals recovered after day 51 has been more systematic.

Figure 2c shows the time evolution of the infection probability rates which are defined by

$$P_e = k_e \frac{U(t)}{S_0} S(t) \tag{3.8}$$

and

$$P_i = k_i \frac{I(t)}{S_0} S(t), \tag{3.9}$$

where $S_0$ is the maximum number of susceptible individuals. The maximum of the infection probability rate of unreported and infected individuals takes place between days 47 and 52 (16–21 April ) and on 60–65 (29 April–4 May), respectively. Therefore, the overall maximum of the infection probability rate ranged

between days of 47–65. These two parameters are reasonable measures of the intensity of daily infection and are proposed on the basis that the zero-order approximation of the joint probability of infection between susceptible and unreported or infected individuals is proportional to the product of the probabilities of being susceptible and infected or unreported.

## 3.3. Domestic confinement and social distance end

The projection of the impact of domestic-confinement end on the evolution of the epidemic assumes that a certain number of individuals returns to the public life and that they can be infected by two mechanisms: (i) poor hygienic conditions, and (ii) contact with an unreported individual. Here, we suppose that contact with infected individuals (those that show symptoms) is less due to the use of prescribed preventative medicines. From the number of individuals that return to its economic activity, one may estimate, in a similar form as in the previous section, that one-third of them become susceptible to infection and, therefore, trigger a pandemic outbreak.

This new community of individuals will have the same structure as the original one ($S_2$, $U_2$, $I_2$, $A_2$, $R_2$, $D_2$) and therefore will show a similar dynamics. However, the triggering contact interactions are modified since they should reflect the actual status of the pandemic, that is, the new community becomes infected because of the community already infected. This may be represented by

$$S_2 + V \xrightarrow{k_v} U_2, \tag{3.10}$$

$$S_2 + U \xrightarrow{k_e} U_2 + U, \tag{3.11}$$

$$S_2 + I \xrightarrow{\hat{k}_i} U_2 + I, \tag{3.12}$$

where we will assume $\hat{k}_i \sim k_i/3$. The successive evolution is determined entirely similar as in the original case, that is, by the sequence of interactions

$$U_2 \xrightarrow{\alpha} I_2, \tag{3.13}$$

$$r\,I_2 \xrightarrow{\gamma} R_2, \tag{3.14}$$

$$d\,I \xrightarrow{\delta} D_2, \tag{3.15}$$

$$a\,I \xrightarrow{\epsilon} A_2, \tag{3.16}$$

$$p\,A_2 \xrightarrow{\delta} D_2, \tag{3.17}$$

$$(1-p)\,A_2 \xrightarrow{\gamma} R_2. \tag{3.18}$$

The differential equations corresponding to the last reactions are similar to the set previously considered, equations (2.10)–(2.15), with the respective reaction rates. For this second community, we have assumed no corrections to the rate constants equivalent to those in equation (2.4), but multiplied the left-hand side of the evolution equations for $S_2$, $U_2$ and $I_2$ by the Heaviside step function $\theta(t - t_f)$ centred at the end of the house confinement measures $t_f$.

Figure 3 shows the consequences of changing the ending of house confinement time, according to the model proposed. Figure 3$a$,$b$ corresponds to ending the domestic confinement at day 90 (29 May) for two new susceptible populations $8 \times 10^4$ and $16 \times 10^4$. The impact is relevant in all the populations considered. The more drastic effects are in the number of infected individuals that increase the pandemic's duration by about 30 days. Most important is that the peak of the number of pre-recovered active cases is shifted to day 150 and increased by about 23%. Figure 3$c$,$d$ corresponds to finishing the domestic confinement at day 110 (4 July) for the same number of new susceptible populations. In this case, the duration of the pandemic increases for about 30 days. The peak of the number of pre-recovered active cases is shifted to day 160 but only increased by about 9%. Finally, it is worth noticing that if the original dynamics could be maintained even after the end of domestic confinement, the pandemic's period is about 300 days. If not, in the worst case considered here, the duration of the pandemic extends to 400 days. If a second possible outbreak is considered, the duration may be increased correspondingly. The cumulative number of infected individuals may also increase.

## 3.4. Model actualization

The original estimations of the susceptible population were performed by using the statistical information on the evolution of the pandemic (up to 19 May), as discussed in §3.2. Further evolution of the pandemic

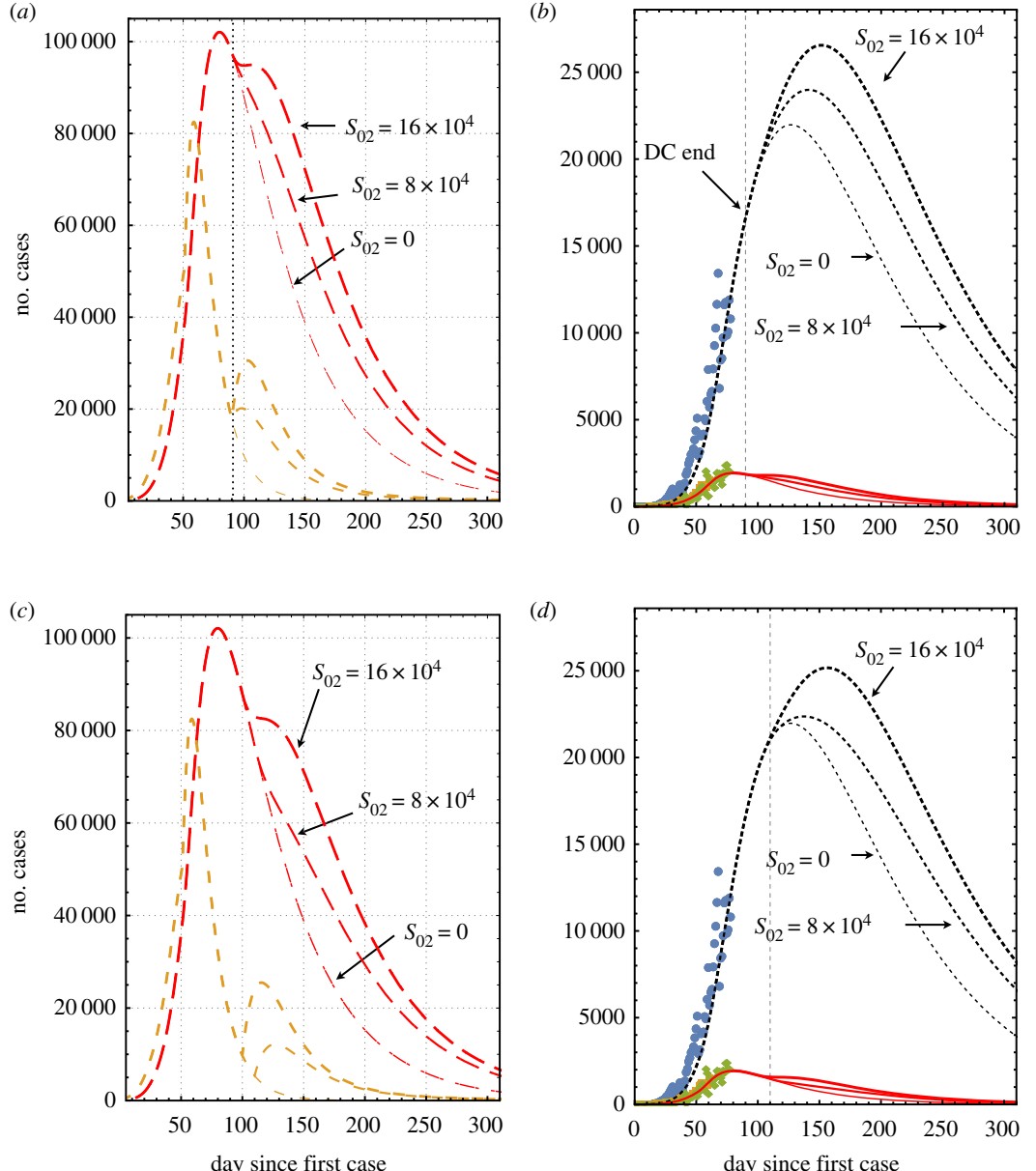

**Figure 3.** Projection of the pandemic evolution after the official estimation to end domestic confinement on days 90 and 110. The line keys are the same as in figure 1. (*a*) Evolution of the pandemic after finishing domestic confinement on day 90 (29 May) assuming new susceptible populations of $8 \times 10^4$ and $16 \times 10^4$. (*b*) Pre-recovered and daily cases associated with (*a*). In (*c*), it is shown the pandemic evolution after finishing domestic confinement at day 110 (4 July). (*d*) Active and daily cases associated with (*c*).

shows (7 August 2020) that the original estimations were insufficient to make an adequate projection. The premature end of the lockdown period at day 90 is leading to new outbreaks that, in quantitative terms, increase dramatically the number of susceptible individuals. Since these outbreaks take place before reaching the maxima of the infected population and daily cases, they are reflected as a direct increase of the initial number of susceptible individuals. By contrast, when the new outbreaks occur after the mentioned peaks then the new susceptible individuals should be incorporated through the mechanism proposed through equations (3.10)–(3.18), as shown in the previous subsection.

The mechanisms that tremendously increased the number of susceptible individuals since the last days of May until the first days of August can be attributed to the impossibility of a strict monitoring of compliance with social distancing, domestic confinement and hygienic measures (like the use of face masks in public spaces and transport). Furthermore, the general confusion on the dispersion, duration and impact of the pandemic in human terms makes it difficult to understand the importance of establishing mobility restrictions among different cities.

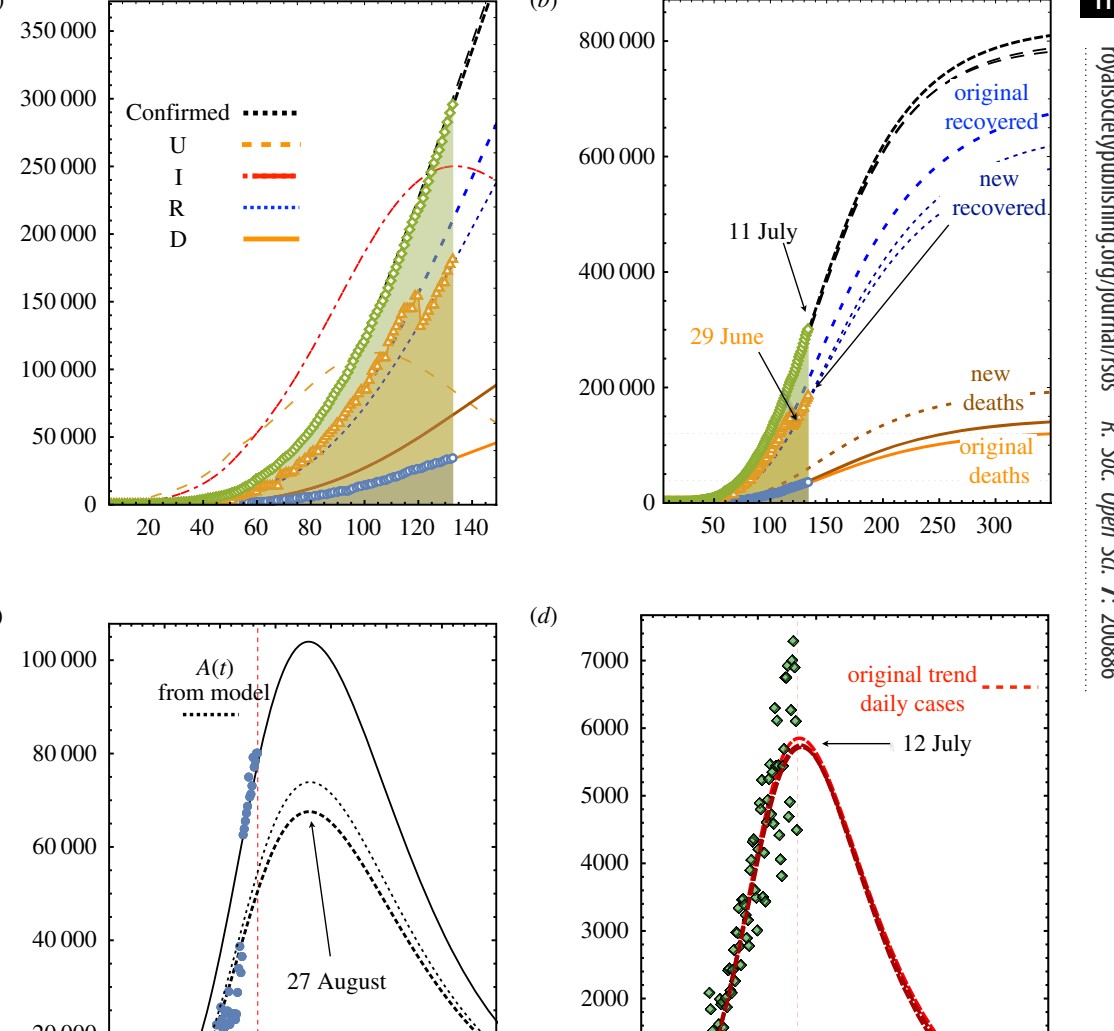

**Figure 4.** Projection of the pandemic evolution with data up to 11 July assuming $S_0 = 850\,000$. (*a*) Evolution of the pandemic after finishing domestic confinement on day 90 (29 May). Orange and dark orange lines below correspond to the number of death trends. (*b*) Evolution of the pandemic up to day 350. (*c*) Active and daily cases associated with (*a*). The solid line corresponds to the new trend in recovered people warranting the fit of the active individuals. The thin dashed line in the middle corresponds to the fit of the death number without warranting the fit of the active individuals. (*d*) Detail of daily cases associated with (*c*).

In this section, we present an actualization of the model to the data reported until 7 August. We performed four fits assuming different initial number of susceptible individuals: $S_0 = 850\,000$ individuals, $S_0 = 1.5$, 3 and 4.5 million individuals. In all cases, the fit criteria is satisfied, although in the last case the prediction of the number of deaths separates appreciably with respect to the data.

Concerning the new fits to the data presented in this subsection, a vital issue occurred during the period going from May to August. At the end of June, the number of recovered individuals reported by the authorities decreased suddenly. On 27 June, the report of recovered individuals was 153 797, and on 29 June, the report was 131 264, a reduction of about 15% [34]. The first actualization was done on 11 July and it is shown in figure 4. The trend of the confirmed and daily cases was accounted for by assuming $S_0 = 850\,000$ individuals. If the restoration of the balance in the total number of individuals infected is done by recalculating only the number of deaths, then this number increases by a factor of 1.86, (see the difference between orange and dark-orange curves in figure 4*a*).

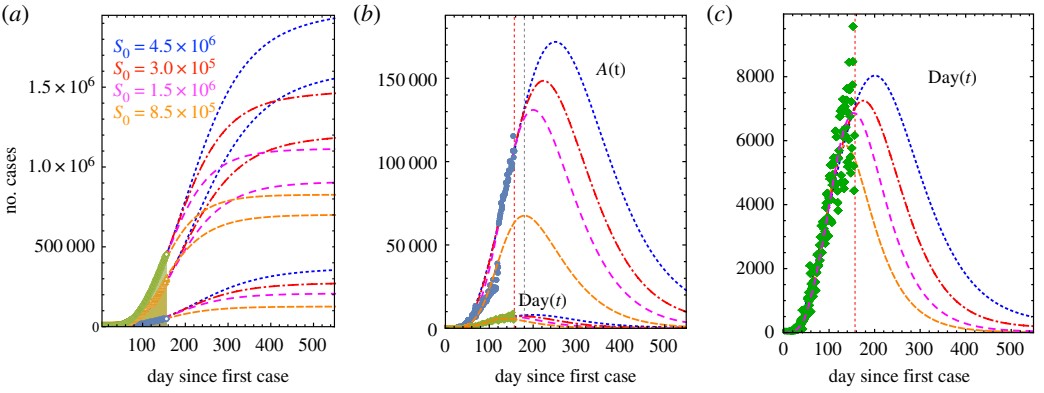

**Figure 5.** Comparison of the projections of the pandemic evolution with data up to 7 August assuming four values of the total susceptible populations: $8.5 \times 10^5$ (orange), $1.5 \times 10^6$ (magenta), $3.0 \times 10^6$ (red) and $4.5 \times 10^6$ (blue). (a) The long-term evolution of the accumulated number of cases, recovered and deaths. (b) The number of pre-recovered active and daily cases. (c) The number of daily cases. Symbols: (a) green (confirmed cases), orange (recovered) and blue (deaths); (b) and (c) blue circles (pre-recovered active) and green (daily cases).

However, this fit is not satisfactory, since the number of pre-recovered active cases is not fitted simultaneously with the other four series of data. It is more adequate to split the difference in the number of recovered individuals between active cases and deaths in such a way that the pre-recovered cases become simultaneously fitted. The resulting increase in the number of deaths is multiplied by a factor of 1.09 since, on day 133, the number of deaths reported was 35 006, see the orange, dark-orange and dashed dark-orange curves in figure 4b. The model prognosis is 38 200, with the assumption of 850 000 susceptible individuals. This factor may seem to be not relevant at the present stage, but after 350 days, the difference becomes about 10 000 individuals. Remarkably, the fit of the daily cases varies by less than 1% for the two methods. The fits of the pre-recovered and daily cases data are shown in figure 4c,d, respectively. These results suggest that the number of daily cases is not reliable enough to take dependable health policies based only on its trends. A more robust quantitative criterion, like the one presented here, is needed.

In figure 5, we present four projections of the pandemic assuming different values of the total number of susceptible individuals with data actualized up to 7 August. The colour code is: orange for $S_0 = 8.5 \times 10^5$, magenta for $S_0 = 1.5 \times 10^6$, red for $S_0 = 3 \times 10^6$ and blue for $S_0 = 4.5 \times 10^4$. The data for confirmed, recovered and death are shown in figure 5a. The orange case was fitted using the original trend of recovered cases to show the large contrast in the behaviour of the pandemic when compared with actualized data. This is more clear in figure 5b where the fit of the pre-recovered and daily cases are also shown. Figure 5c contains the adjustment of daily cases for the assumed susceptible populations. Shown is the two-week shift from peak to subsequent days, associated with increases of 1.5 million in individuals in the susceptible population.

Figure 5 exemplifies the dependence of the transmission rates $k_v$, $k_e$ and $k_i$, on the number of susceptible individuals $S_0$, as indicated in the model's presentation, §2. The values used for the fits are the following: $k_v = 1.2 \times 10^{-4}$, $3.1 \times 10^{-5}$, $3.1 \times 10^{-5}$, $3.1 \times 10^{-5}$ for $S_0 = 8.5 \times 10^5$, $1.5 \times 10^6$, $3 \times 10^6$, $4.5 \times 10^6$, respectively. This means that $k_v$ depends on the proportion of individuals that become recovered or die and not only on the susceptible population. Correspondingly, the other two transmission rates take the values $k_e = 1.4 \times 10^{-7}$, $1.16 \times 10^{-7}$, $5.66 \times 10^{-8}$, $3.5 \times 10^{-8}$ and $k_i = 1.1 \times 10^{-7}$, $1.55 \times 10^{-8}$, $0.95 \times 10^{-9}$, $0.95 \times 10^{-9}$. The other parameters were maintained constant: $\alpha$, $\epsilon = 0.07$, $\gamma = 0.015$ and $\delta = 0.038$. The proportions introduced in equations (2.5)–(2.7) take the values $a = 0.08$, $r = 0.87$ and $g = 0.05$ for the first case, whereas for the remaining three cases the values were $a = 0.15$, $r = 0.79$ and $g = 0.06$.

The fit in this actualization was done by neglecting the economic pressure inattention term introduced for the short-time behaviour described previously. After the time lapse of the inattention initiated at $t_{ep}$, the domestic confinement measures were followed by the population and reduced the effects of the inattention. The consequence was that the present actualization required the introduction of a corrective term inhibiting the transmission rates $k_e$ and $k_i$

$$k_j^0(t) = k_j - \Delta k_j^{di} \theta(t - t_i) - \Delta k_j^{ds} \theta(t - t_{ds}). \tag{3.19}$$

In this case, the second inhibition term initiated at $t_{ds} = 51$, that coincides with the day at which the report in the number of recovered individuals started to be more systematic. The values of the corrections of $\Delta k_e^{di}$ and $\Delta k_e^{ds}$ are $\Delta k_e^{di} \simeq 3.5 \times 10^{-8}$, $2.9 \times 10^{-8}$, $1.4 \times 10^{-8}$, $8.8 \times 10^{-9}\,\mathrm{s}^{-1}$ and $\Delta k_e^{ds} \simeq 2.5 \times 10^{-8}$, $3.6 \times 10^{-9}$, $2.2 \times 10^{-10}$, $2.2 \times 10^{-10}\mathrm{s}^{-1}$ for the total number of susceptible individuals considered above. See [36] for details.

## 4. Conclusion

We have presented and employed a novel model as a conceptual tool to explore the possible fates of the COVID-19 pandemic evolution in the Mexican context. The selection and variation of control parameters in the model allow us to get insights into the system to sustain political decisions on the appropriate time for concluding domestic confinement and social distancing restrictions. The flexibility of the model allows taking into account the specific characteristics of the parameters involved, one composed by the (S, U, I, A, R, D) group and the second one by a particular population that may become susceptible after the end of the domestic confinement measure.

Our model allowed us to perform a very accurate fit of the reported data and, therefore, a reliable projection of the pandemic on the basis that the estimation on the number of susceptible individuals is realistic. We have successfully correlated the inflections of the infection, daily and active cases curves with a decrease of the infection rate after the start of the social distancing measures.

The model also allows predictions of the effect of post-social distancing and house confinement time by considering two elements: (i) poor hygienic conditions, and (ii) contact with an unreported individual. Based on the data published by the Mexican authorities, our first results indicated that the more drastic effects take place in the number of infected individuals by an increase in the duration of the pandemic from roughly 300 days to 400 days in the worst case considered. Additionally, the peak of the number of active cases is shifted to day 150 and increased about 23% if the date of ending social distancing is at day 90. The maximum of the curve of the active case moves towards day 160 when domestic confinement finishes at day 110. As a consequence, the maximum number of pre-recovered active cases increases by about 9%. Since lockdown end was authorized at day 90, before the peak of infected individuals and daily cases occurred, we performed an actualization of the analysis leading to more realistic values for the number of susceptible individuals (1.5, 3 and 4.5 million individuals). These considerations allow us to project three fates of the pandemic that extend over 500 days and more than one million of confirmed cases if no pharmacologic interventions and domestic confinement are further implemented.

Maintaining high hygienic standards (like the permanent use of masks and disinfectants by the individuals) and social distancing measures is the key to successfully control the spread and outbreak of the SARS-CoV-2 after domestic confinement ends. If these measures are not strictly followed and pharmacologic interventions fail, the pandemic duration can last more than a year and a half with a dramatic number of deaths, estimated to be approximately 200 000 or higher.

Data accessibility. Electronic supplementary material are available at the Dryad Digital Repository: https://doi.org/10.5061/dryad.70rxwdbv5 [36].

Competing interests. We declare that we have no competing interests.

Authors' contributions. I.S.-H. carried out program coding and model simulations, conceived the study, carried out the analysis of the results and drafted the manuscript; V.C. conceived the study, carried out the analysis of the results and helped draft the manuscript. All authors gave final approval for publication and agree to be held accountable for the work performed therein.

Acknowledgements. We are grateful to Dr Aldo Ledesma-Durán and Prof. Karo Michaelian for reading the manuscript and sharing valuable comments on its content.

Funding. This work was supported by UNAM-DGAPA under grant no. IN117419.

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
