## [Reviewer comments · Royal Society Open Science]

Review History

RSOS-200886.R0 (Original submission)

Review form: Reviewer 1

Is the manuscript scientifically sound in its present form?

No

Are the interpretations and conclusions justified by the results?

Yes

Is the language acceptable?

Yes

Do you have any ethical concerns with this paper?

No

Have you any concerns about statistical analyses in this paper?

No

Recommendation?

Major revision is needed (please make suggestions in comments)

Comments to the Author(s)

The authors use a compartment model to explore the spread of SARS-Cov-2 infections across Mexico, with time-dependent infection rates. In particular, the authors investigate the effect of social distancing and house confinement measures on the evolution of epidemics in Mexico.

I am afraid that the authors are not aware of the large number of SIR/SEIR models published over the past few months in the context of Covid-19 spread. These references are completely missing from the Introduction (where the authors reference only 2 studies [23] and [15]). And this lack of knowledge made the authors to claim that the novelty of their study is the time-dependence of infection rates due to restrictions being imposed/removed.

But there are many studies that investigate the effect of covid-19 restrictions (in one way or another). See, for example,

[https://www.thelancet.com/journals/laninf/article/PIIS1473-3099\(20\)30144-4/fulltext](https://www.thelancet.com/journals/laninf/article/PIIS1473-3099(20)30144-4/fulltext)

[https://www.thelancet.com/journals/lanpub/article/PIIS2468-2667\(20\)30073-6/fulltext](https://www.thelancet.com/journals/lanpub/article/PIIS2468-2667(20)30073-6/fulltext)

The results of this study (as well as the derivation of the new model presented in this manuscript) needs to be discussed in the context of the published literature (for other epidemics, as well as the current Covid-19 epidemic).

I would say that the novelty of this study is the application of this model to Mexico data... But I need to see a stronger discussion of the various SEIR models used for Covid-19, and how the current model compares with published models that also investigate the effect of various restrictions.

A few other minor issues:

First the authors use the word "dispersion" when they refer to SARS-Cov-2 spread (they use it even in the title of the manuscript). This is misleading since their model does not incorporate spatial dispersal terms. They should replace the word "dispersion" with the word "spread".

Page 5 lines 17-19: the letters "Âã" appear randomly inside some words

Review form: Reviewer 2

Is the manuscript scientifically sound in its present form?

Yes

Are the interpretations and conclusions justified by the results?

Yes

Is the language acceptable?

Yes

Do you have any ethical concerns with this paper?

No

Have you any concerns about statistical analyses in this paper?

No

Recommendation?

Major revision is needed (please make suggestions in comments)

Comments to the Author(s)

Please see the attached files (Appendix A).

Decision letter (RSOS-200886.R0)

Dear Dr Santamaria-Holek,

The editors assigned to your paper ("Possible fates of the dispersion of SARS-COV-2 in the Mexican context") have now received comments from reviewers. We would like you to revise your paper in accordance with the referee and Associate Editor suggestions which can be found below (not including confidential reports to the Editor). Please note this decision does not guarantee eventual acceptance.

Please submit a copy of your revised paper before 13-Aug-2020. Please note that the revision deadline will expire at 00.00am on this date. If we do not hear from you within this time then it will be assumed that the paper has been withdrawn. In exceptional circumstances, extensions may be possible if agreed with the Editorial Office in advance. We do not allow multiple rounds of revision so we urge you to make every effort to fully address all of the comments at this stage. If deemed necessary by the Editors, your manuscript will be sent back to one or more of the original reviewers for assessment. If the original reviewers are not available, we may invite new reviewers.

- Data accessibility

It is a condition of publication that all supporting data are made available either as supplementary information or preferably in a suitable permanent repository. The data accessibility section should state where the article's supporting data can be accessed. This section should also include details, where possible of where to access other relevant research materials such as statistical tools, protocols, software etc can be accessed. If the data have been deposited in

an external repository this section should list the database, accession number and link to the DOI for all data from the article that have been made publicly available. Data sets that have been deposited in an external repository and have a DOI should also be appropriately cited in the manuscript and included in the reference list.

<http://datadryad.org/submit?journalID=RSOS&manu=RSOS-200886>

- **Competing interests**

- **Authors' contributions**

- **Acknowledgements**

- **Funding statement**

on behalf of Professor Tim Rogers (Associate Editor) and Mark Chaplain (Subject Editor)
openscience@royalsociety.org

Reviewers' Comments to Author:

Reviewer: 1

Comments to the Author(s)

The authors use a compartment model to explore the spread of SARS-Cov-2 infections across Mexico, with time-dependent infection rates. In particular, the authors investigate the effect of social distancing and house confinement measures on the evolution of epidemics in Mexico.

I am afraid that the authors are not aware of the large number of SIR/SEIR models published over the past few months in the context of Covid-19 spread. These references are completely missing from the Introduction (where the authors reference only 2 studies [23] and [15]). And this lack of knowledge made the authors to claim that the novelty of their study is the time-dependence of infection rates due to restrictions being imposed/removed.

But there are many studies that investigate the effect of covid-19 restrictions (in one way or another). See, for example,

[https://www.thelancet.com/journals/laninf/article/PIIS1473-3099\(20\)30144-4/fulltext](https://www.thelancet.com/journals/laninf/article/PIIS1473-3099(20)30144-4/fulltext)

[https://www.thelancet.com/journals/lanpub/article/PIIS2468-2667\(20\)30073-6/fulltext](https://www.thelancet.com/journals/lanpub/article/PIIS2468-2667(20)30073-6/fulltext)

The results of this study (as well as the derivation of the new model presented in this manuscript) needs to be discussed in the context of the published literature (for other epidemics, as well as the current Covid-19 epidemic).

I would say that the novelty of this study is the application of this model to Mexico data... But I need to see a stronger discussion of the various SEIR models used for Covid-19, and how the current model compares with published models that also investigate the effect of various restrictions.

A few other minor issues:

First the authors use the word "dispersion" when they refer to SARS-Cov-2 spread (they use it even in the title of the manuscript). This is misleading since their model does not incorporate spatial dispersal terms. They should replace the word "dispersion" with the word "spread".

Page 5 lines 17-19: the letters "Âã" appear randomly inside some words

Reviewer: 2

Comments to the Author(s)

Please see the attached files.

Author's Response to Decision Letter for (RSOS-200886.R0)

See Appendix B.

RSOS-200886.R1 (Revision)

Review form: Reviewer 1

Is the manuscript scientifically sound in its present form?

Yes

Are the interpretations and conclusions justified by the results?

Yes

Is the language acceptable?

Yes

Do you have any ethical concerns with this paper?

No

Have you any concerns about statistical analyses in this paper?

No

Recommendation?

Accept as is

Comments to the Author(s)

The authors have addressed my previous comments.

Review form: Reviewer 2

Is the manuscript scientifically sound in its present form?

Yes

Are the interpretations and conclusions justified by the results?

Yes

Is the language acceptable?

Yes

Do you have any ethical concerns with this paper?

No

Have you any concerns about statistical analyses in this paper?

No

Recommendation?

Accept as is

Comments to the Author(s)

Dear authors, thank you for your responses to my comments. I consider that your answers are clear and well written. I recommend the publication of this work.

Decision letter (RSOS-200886.R1)

Dear Dr Santamaria-Holek,

It is a pleasure to accept your manuscript entitled "Possible fates of the spread of SARS-COV-2 in the Mexican context" in its current form for publication in Royal Society Open Science. The comments of the reviewer(s) who reviewed your manuscript are included at the foot of this letter.

COVID-19 rapid publication process:

We are taking steps to expedite the publication of research relevant to the pandemic. If you wish, you can opt to have your paper published as soon as it is ready, rather than waiting for it to be published the scheduled Wednesday.

This means your paper will not be included in the weekly media round-up which the Society sends to journalists ahead of publication. However, it will still appear in the COVID-19 Publishing Collection which journalists will be directed to each week (<https://royalsocietypublishing.org/topic/special-collections/novel-coronavirus-outbreak>).

If you wish to have your paper considered for immediate publication, or to discuss further, please notify openscience_proofs@royalsociety.org and press@royalsociety.org when you respond to this email.

on behalf of Professor Tim Rogers (Associate Editor) and Mark Chaplain (Subject Editor)
openscience@royalsociety.org

Reviewer comments to Author:
Reviewer: 2

Comments to the Author(s)

Dear authors, thank you for your responses to my comments. I consider that your answers are clear and well written. I recommend the publication of this work.

Reviewer: 1

Comments to the Author(s)

The authors have addressed my previous comments.

Appendix A

Review of "Possible fates of the dispersion of SARS-COV-2 in the Mexican context"

by Santamaria-Holek, Ivan and Castaño, Victor Manuel

July 15, 2020

The authors propose a compartmental model to study the dynamics of the SARS-CoV-2 epidemic in the Mexican context. The model proposed is an extension of the classical SEIR epidemic model. This study is interesting and well-motivated. However, there are several important points that need attention:

Major comments

1. Usually, in epidemiology, when a healthy individual who is vulnerable to contracting a disease makes potentially disease-transmitting contact, that individual becomes exposed. Exposed individuals are typically not infectious [1]. However, in this work the authors assume Exposed individuals are always infectious. For me, the compartment E that the authors called the exposed, it is actually more a pre-symptomatic or asymptomatic infectious compartment and the actual latent period is ignored (because infected individuals are immediately infectious). Thus, using the named *exposed* is not precise according to the model formulation.
2. In the model formulation, the definition of an active case should be explained in more detail. For example, why an active case is always infected but not infectious? How can an infected individuals manifest active symptoms (and therefore be in the active class) but not to be infectious? In other words, how an infected individuals losses its infectiousness but is still infected? When the active cases are not infectious they are actually irrelevant for the transmission of the disease, but I believe this is not always the case. So, further explanation is needed. Related to this, in page 4, line 25, I think the definition is for ϵ^{-1} and not for ϵ .
3. Again, in the model formulation, the definition of the infection rates k_j is not clear. Usually, the effective contact rate is the product of the per capita contacts per unit time, times the probability of infection per contact. For a flu-like disease like COVID-19, the Kermack-McKendrick type of model (as used in this study) is appropriate. However, the introduction of the infection rate for the free-living virus makes the interpretation of the equations ambiguous. What is exactly k_v ? How is it defined? If infections from the free-living virus V are accounted for, what is the meaning of the other two contact rates k_e and k_i ? How would they be measured? How would k_v be measured?
4. The explicit introduction of the virus V for modeling flu-like diseases is interesting. However, in the model proposed by the authors I did not see any equation for V (does that mean that V is constant?). The amount of free-living virus (SARS-CoV-2) in the environment depends on the shedding by infectious individuals and it is not constant. So, an inclusion of a dynamic equation for V is more realistic and it is necessary in my opinion (the inclusion of the virus V should be done carefully from a modeling perspective). Other authors have considered a compartment for the free-living virus V but they consider an explicit equation for V [3].

5. Page 3. Lines 49-53. The authors wrote that there is a lack of data on the number of exposed individuals that do not develop symptoms and they assume $q \approx 1$. The authors even mention that this assumption is drastic and should be made to avoid speculations on the exposed. However, several studies have suggested that approximately 80% of SARS-CoV-2 infections show mild or no symptoms (see page 5 in [2] and the references there in). Therefore, the value of the parameter q needs to be modified to reflect this fact. This is important because in other case the infections caused by asymptomatic individuals are underestimated.
6. Page 5, lines 10-14. I do not think the estimation $S_0 = 305$ thousand individuals is correct (if the total Mexican population is more than 125 million people, there is no way that only 305 thousand is the number of susceptibles). Maybe, that estimation is for I_0 . In page 5, line 57, the authors even use $S_0 = 187$ thousand (the results are in Figure 1b), so can the authors explain in simple terms why are they using such initial populations?
7. Page 6. Lines 52-55. This comment is related with the comment 4. Do the authors have a reference to postulate that the effective virulence factor $k_v V$ has a value in the range 10^{-6} - 10^{-5} ? Moreover, the virulence is by definition the severity or harmfulness of a disease or poison [1]. So, it does not look precise to my, called the variable V the virulence. The expression $k_v V$ contains the information on object-to-person transmission (and not virulence). In summary, the rate k_v and the parameter V are defined ambiguously.

Minor comments

1. Sometimes the authors write SARS-COV-2 and other times they write SARS-CoV-2. Please be consistent.
2. There are typo mistakes in page 4 lines 17 and 19.
3. Typo mistake in page 5 line 51 (details instead of deatils).

I hope these comments might be helpful for the authors. If the authors can correct accurately the above issues, I recommend the publication of this work.

References

- [1] Martcheva, M. (2015). An introduction to mathematical epidemiology (Vol. 61). New York: Springer.
- [2] Ngonghala, C. N., Iboi, E., Eikenberry, S., Scotch, M., MacIntyre, C. R., Bonds, M. H., & Gumel, A. B. (2020). Mathematical assessment of the impact of non-pharmaceutical interventions on curtailing the 2019 novel Coronavirus. *Mathematical Biosciences*, 108364.
- [3] Saldaña, F., Flores-Arguedas, H., Camacho-Gutierrez, J. A., & Barradas, I. (2020). Modeling the transmission dynamics and the impact of the control interventions for the COVID-19 epidemic outbreak.

Appendix B

August 10, 2020

Dear Professors Tim Rogers and Mark Chaplain

Journal of the Royal Society Open Science

We want to acknowledge sending us the reviews of our paper "Possible fates of the spread of SARS-CoV-2 in the Mexican context" with identification number ID RSOS-200886, and giving us the opportunity to make revisions of the contents according to the Reviewers' suggestions. We also acknowledge the Reviewers for taking the time to carefully read the manuscript and make valuable comments on it.

Following the Reviewer's suggestions, we have improved the introduction of our manuscript by adding recent references on the subject and discussing the actual context on the modeling of the spread of the SARS-CoV-2 virus. We improved the model's presentation section by explaining more thoroughly all its ingredients as well as including an actualization section to discuss the current situation of the evolution of the COVID-19 disease in Mexico.

In what follows, we include the answers to the Reviewer's comments and questions, accompanied by a short description of the amendments we have made in the manuscript. Finally, we also provide a highlighted version of the paper indicating in blue color the changes performed.

In what follows, we include the answers to the Reviewer's comments and questions, accompanied by a short description of the amendments we have made in the manuscript. Finally, we also provide a highlighted version of the article indicating in blue color the changes performed. We hope that this new version could be suitable for publication in the Journal of the Royal Society Open Science.

Sincerely Yours,

Iván Santamaría-Holek and Víctor Castaño

Reviewers' Comments to Author:

Reviewer: 1

Comments to the Author(s)

The authors use a compartment model to explore the spread of SARS-Cov-2 infections across Mexico, with time-dependent infection rates. In particular, the authors investigate the effect of social distancing and house confinement measures on the evolution of epidemics in Mexico.

I am afraid that the authors are not aware of the large number of SIR/SEIR models published over the past few months in the context of Covid-19 spread. These references are completely missing from the Introduction (where the authors reference only 2 studies [23] and [15]). And this lack of knowledge made the authors to claim that the novelty of their study is the time-dependence of infection rates due to restrictions being imposed/removed.

But there are many studies that investigate the effect of covid-19 restrictions (in one way or another). See, for example,

[https://www.thelancet.com/journals/laninf/article/PIIS1473-3099\(20\)30144-4/full-text](https://www.thelancet.com/journals/laninf/article/PIIS1473-3099(20)30144-4/full-text)

[https://www.thelancet.com/journals/lanpub/article/PIIS2468-2667\(20\)30073-6/full-text](https://www.thelancet.com/journals/lanpub/article/PIIS2468-2667(20)30073-6/full-text)

The results of this study (as well as the derivation of the new model presented in this manuscript) needs to be discussed in the context of the published literature (for other epidemics, as well as the current Covid-19 epidemic).

We agree with the Reviewer and thank her/him for sharing these references that are important for our work's presentation and rationale. We have added them in the references [16,17] and discuss their contribution in the main text. We also added two more recent references that also help to contextualize our model within the COVID context [19,20]. We have searched for the relevant literature with the aim to maintain a relatively short number of references.

Detailed analysis of these models and a discussion on the similarities and differences with our work have been included in various paragraphs of the new version of our manuscript. We have performed a more detailed search of specialized literature, relevant to our article, in particular research articles that have appeared in the last three months.

Adjustments in the manuscript: We added commentaries on the research cited by the Reviewers and other opportune for the better presentation of the work in paragraphs 3 and 4 of page 2; included the references and cited them in the manuscript where they are opportune.

I would say that the novelty of this study is the application of this model to Mexico data... But I need to see a stronger discussion of the various SEIR models used for Covid-19, and how the current model compares with published models that also investigate the effect of various restrictions.

We believe that our work's new presentation clarifies our model's novelties, besides the specific application to Mexico's data. The critical point of the Mexican context is associated with two main issues: one is the lack of information needed for traditional approaches based in compartment models, in particular of the exposed sector due to the low quantity of detection tests applied by the authorities. The second point is the delay of incorporating information about the state of the different sectors, a fact that introduces additional delays in the behavior of the different compartments considered.

A few other minor issues:

First the authors use the word "dispersion" when they refer to SARS-Cov-2 spread (they use it even in the title of the manuscript). This is misleading since their model does not incorporate spatial dispersal terms. They should replace the word "dispersion" with the word "spread".

The replacement was made, as suggested.

Page 5 lines 17-19: the letters "Ã" appear randomly inside some words

This has been corrected in the whole text.

Reviewer: 2

Comments to the Author(s)

The authors propose a compartmental model to study the dynamics of the SARS-CoV-2 epidemic in the Mexican context. The model proposed is an extension of the classical SEIR epidemic model. This study is interesting and well-motivated. However, there are several important points that need attention:

Major comments

1. Usually, in epidemiology, when a healthy individual who is vulnerable to contracting a disease makes potentially disease-transmitting contact, that individual becomes exposed. Exposed individuals are typically not infectious [1]. However, in this work the authors assume Exposed individuals are always infectious. For me, the compartment E that the authors called the exposed, it is actually more a pre-symptomatic or asymptomatic infectious compartment and the actual latent period is ignored (because infected individuals are immediately infectious). Thus, using the named *exposed* is not precise according to the model formulation.

We agree with this comment. In classical epidemiological models and, therefore, their basic definitions, the Exposed individuals, usually represented by E, are not infectious and, therefore, they do not contribute to the infection force. We realize now that our use of this term and the notation employed in the manuscript introduces confusion in our presentation of the model. We corrected this in the new version of the manuscript by being more transparent in defining the different sectors.

Our model aims to reproduce the data reported by the health authorities and, to achieve this objective, we considered necessary to introduce a compartment (E) in which we have taken into account all the non-reported individuals (exposed, asymptomatic and pre-symptomatic) that contribute to the spread of the virus and therefore should enter in the infection force term. We now see that it is more convenient to distinguish this compartment from the Exposed individuals' compartment in the traditional sense and, therefore, in the new version of the manuscript, we have changed the notation to avoid confusion, along with a more careful explanation of this point.

In this respect, and following the recommendation raised by Reviewer 1 on searching for more adequate references to our work, the new version of the manuscript includes the reference: *Infec. Dis. Mod.* 5 (2020) 323. In this work, an approach similar to ours to account for unreported infectious and non-infectious individuals was adopted. We now include the corresponding reference to this work in the presentation of the model.

In the new version of the manuscript, we have removed the notation and the reference to Exposed individuals and introduced, following the notation of the new Ref. [20], the Unreported (exposed, asymptomatic, and pre-symptomatic) individuals.

Adjustments in the manuscript: This point and the following ones raised by the Reviewer 2 required extensive changes in the presentation of the manuscript with the aim to improve the presentation of the model, since it differs appreciably from traditional SEIRD models.

This means that we have almost modified the whole section 2 to improve the presentation of the model. We have changed the capital letter E by U and changed the word “exposed” by “unreported” along the manuscript and the formulas. We also considered adequate to separate the formulation of the first 3 reactions to better define their assumptions and implications, as well as the parameters entering on them.

2. In the model formulation, the definition of an active case should be explained in more detail. For example, why an active case is always infected but not infectious? How can an infected individuals manifest active symptoms (and therefore be in the active class) but not to be infectious? In other words, how an infected individuals losses its infectiousness but is still infected? When the active cases are not infectious they are actually irrelevant for the transmission of the disease, but I believe this is not always the case. So, further explanation is needed. Rela-

-1

ted to this, in page 4, line 25, I think the definition is for ϵ and not for ϵ .

The compartment A accounts for the difference of the cumulative number of individuals minus the recovered ones minus the dead ones, as defined in the manuscript (see also <https://www.worldometers.info/coronavirus/#countries>). We have associated this compartment to "active" individuals because, as far as we know, there is no reference to this compartment in the literature.

After we submitted the manuscript, the WHO published the following information concerning the criteria for releasing COVID-19 patients from isolation. We cite the first

paragraph of the original text taken from <https://www.who.int/news-room/commentaries/detail/criteria-for-releasing-covid-19-patients-from-isolation>:

“On 27 May 2020, WHO published updated interim guidance on the clinical management of COVID-19,1,2 and provided updated recommendations on the criteria for discharging patients from isolation. The updated criteria reflect recent findings that patients whose symptoms have resolved may still test positive for the COVID-19 virus (SARS-CoV-2) by RT-PCR for many weeks. Despite this positive test result, these patients are not likely to be infectious and therefore are unlikely to be able to transmit the virus to another person.”

Given this, we have decided to redefine the compartment A as that accounting for the pre-Recovered active individuals. These are those individuals whose symptoms have resolved but may still test positive for the COVID. That is, they are still infected but are (probably) not infectious.

A final comment on this point is that the period that Mexican health authorities determine for active individuals is 14 days. After this period, those individuals are no longer considered active. Before introducing the compartment A, we were able to fit the cumulative, recovered, and death data simultaneously, but we were unable to fit the daily cases' dynamics. Compartment A introduced a secondary delay characterized by the time scale $1/\epsilon$ ($= 14$ days) that allowed us to fit the daily cases and the data corresponding to the pre-recovered ones (the difference of the cumulative number of individuals minus the recovered ones minus the death ones) simultaneously.

Adjustments in the manuscript: We added the previous commentaries in the paragraph before Eqs. (2.5)-(2.7) on page 5. We have redefined the compartment A as a pre-recovered and changed the word “active” by “pre-recovered active” along the manuscript. We also corrected the exponent in the definitions of ϵ and α .

3. Again, in the model formulation, the definition of the infection rates k_j is not clear. Usually, the effective contact rate is the product of the per capita contacts per unit time, times the probability of infection per contact. For a flu-like disease like COVID-19, the Kermack- McKendrick type of model (as used in this study) is appropriate. However, the introduction of the infection rate for the free-living virus makes the interpretation of the equations ambiguous. What is exactly k_j ? How is

it defined? If infections from the free-living virus V are accounted for, what is the meaning of the other two contact rates k_e and k_i ? How would they be measured?

How would k_v be measured?

These are two interesting points that we improved in the new presentation of the model.

The parameters k_e and k_i were used by assuming they are proportional to a contact probability per unit time and, implicitly, inversely proportional to the number of susceptible individuals. In the new version of the manuscript, we explicitly mention this dependence to avoid confusion under Eqs. (2.2) and (2.3).

We avoided the usual presentation of compartmental models since we attempt to approach the problem directly, although less formally in the mathematical sense. The essential difference is that the values of parameters, inferred a priori by following the usual considerations on the total country population, clinical data, or taken from the literature, were successively bettered through the simultaneous fittings of the accumulated, recovered, dead, daily, and active cases.

The introduction of a constant term kv^*V in the evolution equation for the susceptible population is equivalent to the assumption in reference: *Infec. Dis. Mod.* 5 (2020) 323. The authors of that innovative work introduce an initial period having an exponential increase in the cumulative number of cases before switching-in their SEIR-like model. The results are similar to ours, the “translation” of the non-linear dynamics in time.

Therefore, the coefficient kv^*V is an average rate of contagions that incorporates human and non-human mediated contacts with the virus and it is also a decreasing function of the number of susceptible individuals. The simple interpretation of this contribution allowed us to fit the early evolution of the pandemic, which is nearly exponential and precedes the evolution period dominated by the non-linear nature of human contacts.

We agree, however, that the presentation of the parameter kv^*V was confusing in the original version of the manuscript since we have divided it into two contributions, the rate of (free) virus infection and the “quantity” of the (free) virus leading to contagions.

In the new version of the manuscript, we have simplified the justification, making it more direct and clear, see point 7, and referencing the article *Infec. Dis. Mod.* 5 (2020) 323.

Adjustments in the manuscript: In the new version of the manuscript, we explicitly mention the dependence of the transmission rates k_e and k_i on the number of susceptible individuals and the contact probability per day after Eqs. (2.2) and (2.3) in page 4. In addition, the justification of the introduction of Eq. (2.1) involving the parameters k_v and V was improved in the last paragraph of page 3 making reference to previous models that followed similar [20] and different approaches [19]. We also improved the justification and objective of the model at the beginning of the section 2.

4. The explicit introduction of the virus V for modeling flu-like diseases is interesting. However, in the model proposed by the authors I did not see any equation for V (does that mean that V is constant?). The amount of free-living virus (SARS-CoV-2) in the environment depends on the shedding by infectious individuals and it is not constant. So, an inclusion of a dynamic equation for V is more realistic and it is necessary in my opinion (the inclusion of the virus V should be done carefully from a modeling perspective). Other authors have considered a compartment for the free-living virus V but they consider an explicit equation for V [3].

In our model, the free-virus' influence is incorporated implicitly in the value of the rate coefficients k_e and k_i , and k_v . In the previous point, we have already explained that we considered the first term of the infection force as constant because it allows us to account for the early evolution of the pandemic.

The postulation of an evolution equation for V that depends on the other compartments implicitly establishes an equivalence of contagions and dynamics among the free-virus and individuals with the contagions among different sectors of individuals. We consider that this is not convenient, and therefore we have characterized the influence of the free-virus through an average factor responsible, in part, of the initial exponential increase in cases.

The postulation of the free-virus dynamics is interesting since it may provide one with a more detailed vision for the contagions. However, it has the disadvantage that requires new parameters for its characterization.

In our approximation, the free-virus presence is established only through a single relation between the virus and the susceptible population and, therefore, it can be measured directly in terms of the number of direct contacts with free-human sources. The

excellent fit of the data and the similarity with previous reports in the literature make us ever more confident of the model's validity.

Adjustments in the manuscript: The adjustments of this point go together with those of point 3. In particular, we mention the possibility of using an evolution equation for the free-virus in the second paragraph of page 4, in the discussion of the presentation of Eq. (2.1).

5. Page 3. Lines 49-53. The authors wrote that there is a lack of data on the number of exposed individuals that do not develop symptoms and they assume $q \approx 1$. The authors even mention that this assumption is drastic and should be made to avoid speculations on the exposed. However, several studies have suggested that approximately 80% of SARS-CoV-2 infections show mild or no symptoms (see page 5 in [2] and the references there in). Therefore, the value of the parameter q needs to be modified to reflect this fact. This is important because in other case the infections caused by asymptomatic individuals are underestimated.

In point 1, we have explained that our compartment E (now U) does not correspond to the usual definition of Exposed individuals, but the number of Unreported individuals (exposed, pre-symptomatic and asymptomatic). Since the compartment represents the unreported individuals, it is, by definition, not considered in the data reported by the authorities. The consequence is that the Unreported individuals who became recovered do not appear in the Recovered or Active individuals' statistics. Therefore, our model assumes that the Unreported individuals only leave their compartment when they pass to the (reported) Infected compartment and that the Recovered individuals, that are reported by necessity, only came from the compartment I. These considerations are consistent with the public data and self-consistent. They imply that the parameter "q" measures the quantity of reported pre-symptomatic and asymptomatic individuals irrespective of the severity of their condition, and not the proportion of Exposed individuals that may develop, or not, severe conditions.

It is clear that the necessity to incorporate the Unreported compartment, with its implications, can be reduced or eliminated by the massive application of tests that may identify the real proportions of pre-symptomatic and asymptomatic cases before they become individuals with mild or severe condition. Unfortunately, however, this is not the present case.

Adjustments in the manuscript: To avoid confusion by the inclusion of the parameters “q” and “beta” of the first version of the manuscript, we have simplified the equation (2.4) in page 5 by removing the second term. This change do not affects the results of the work but simplifies considerably the presentation and implications of the existence of the unreported compartment for case of study.

6. Page 5, lines 10-14. I do not think the estimation $S_0 = 305$ thousand individuals is correct (if the total Mexican population is more than 125 million people, there is no way that only 305 thousand is the number of susceptibles). Maybe, that estimation is for I_0 . In page 5, line 57, the authors even use $S_0 = 187$ thousand (the results are in Figure 1b), so can the authors explain in simple terms why are they using such initial populations?

The first estimation was of the number of susceptible individuals was obtained using statistical data from some European countries and taking into account that the pandemic could be controlled in urban centers. Through time, this estimation under-represented the situation in Mexico and other countries. The second (more inadequate) estimation that we used to fit the data until May 19 assumed that the authorities' statements about the tendencies of several factors indicated that the maximum of the infected individuals should occur about the middle days of May.

We know now that the pandemic is going out from the cities and increases in smaller towns. There is a broad and aggressive dispersion of the virus in the country. After day 90, the decision to open several economic activities lead to a sudden increase in the number of Susceptible individuals. This fact, accompanied by the economic pressure circumstances over the population, further increased the number of cases in a few weeks.

In the new version of the manuscript, we keep the original estimation but incorporated a new subsection we named Model Actualization, in which the fit of the data is accomplished until the first days of August. In this new subsection, we show several things: first, we clarified the dependence of the coefficients k_e and k_i on the total number of susceptible individuals. Second, that our initial estimation (based on statistical data from other countries) was insufficient because it was based on statistical information of other countries in which the pandemic did not finished. Third, we present four fits consistent with the number of pre-Recovered active individuals and the daily cases. The smaller

increase (850 thousand individuals) was done with data up to July 11 and it is consistent with the low value of the pre-recovered cases data, that is the trend shown before the date reported for recovered individuals was "corrected" at the beginning of June. The other three cases (1.5, 3 and 4.5 million individuals) correspond to the fit of the data with corrected data of Recovered individuals (the authorities diminished the amount by about 15%) and are consistent with the average values of the active and daily cases. They show how the increase on the number of susceptible individuals shifts the peak of the pre-recovered and daily cases by about two weeks.

Adjustments in the manuscript: We have improved the discussion on the first estimation of the initial susceptible population by adding, before section 3 in page 6, comments of the previous response. Additionally, we have actualized the model projections up to August 7 by introducing the subsection "3d) Model actualization" on page 11-13. The development of the actualization is also used to illustrate the dependence of the coefficients k_e and k_i on the total number of susceptible individuals. Two new figure sets, 4 and 5, are also included. Finally, we moved the paragraph discussing the time dependence of k_e and k_i from the model presentation section to the results section on page 7, and included the formula (3.19).

7. Page 6. Lines 52-55. This comment is related with the comment 4. Do the authors have a reference to postulate that the effective virulence factor k_V has a

$$-6 \quad -5$$

value in the range 10^{-6} to 10^{-5} ? Moreover, the virulence is by definition the severity or harmfulness of a disease or poison [1]. So, it does not look precise to me, called the variable V the virulence. The expression k_V contains the information

on object-to-person transmission (and not virulence). In summary, the rate k and the parameter V are defined ambiguously.

The use of the word virulence was removed in the new version of the manuscript.

We explained the interpretation of the parameter k_v in point 4: it refers to the number of contacts per number of susceptible individuals with (predominant but not exclusive) no-human agents at the early stages of the pandemic. The parameter is important because the number of individuals that may have direct contact with no-human virus sources is much lower than the total population of a country. The consequence of this approach is that the early exponential growth of the pandemic only dominates the dy-

namics during the initial days. After this initial period, about 20 days, human contacts' nonlinear dynamics start to dominate the behavior. When the nonlinear dynamics are near switch-off, the final stages are also led by the exponential term, but the time variation is weak.

Adjustments in the manuscript: The adjustments on this point go along with those of point 3 and 4. The values for the coefficient k_v are now explicitly reported on page 13.

Minor comments

1. Sometimes the authors write SARS-COV-2 and other times they write SARS-CoV-2. Please be consistent.

We corrected this typo.

2. There are typo mistakes in page 4 lines 17 and 19.

We corrected these typos.

3. Typo mistake in page 5 line 51 (details instead of deatils).

We corrected this typo.

I hope these comments might be helpful for the authors. If the authors can correct accurately the above issues, I recommend the publication of this work.

We thank the Reviewer 2 by looking so carefully to our model and make pertinent commentaries on presentation and significance. All these comments were very useful to improve the presentation of the model and rationalize the results. We hope that, after the changes done, the new version of the manuscript is now acceptable for publication.